# Prognostic Impact of Microscopic Extra-Thyroidal Extension (mETE) on Disease Free Survival in Patients with Papillary Thyroid Carcinoma (PTC)

**DOI:** 10.3390/cancers14112591

**Published:** 2022-05-24

**Authors:** Nadia Bouzehouane, Pascal Roy, Myriam Decaussin-Petrucci, Mireille Bertholon-Grégoire, Chantal Bully, Agnès Perrin, Helene Lasolle, Jean-Christophe Lifante, Françoise Borson-Chazot, Claire Bournaud

**Affiliations:** 1Fédération d’Endocrinologie, Hospices Civils de Lyon, Groupement Hospitalier Est, 69500 Bron, France; helene.lasolle@chu-lyon.fr (H.L.); francoise.borson-chazot@chu-lyon.fr (F.B.-C.); 2Université de Lyon, 69000 Lyon, France; pascal.roy@chu-lyon.fr; 3Université Claude Bernard-Lyon 1, 69100 Villeurbanne, France; 4Service de Biostatistique et Bioinformatique, Pôle Santé Publique, Hospices Civils de Lyon, 69373 Lyon, France; 5Laboratoire de Biométrie et Biologie Évolutive, CNRS UMR 5558, 69100 Villeurbanne, France; 6Health Services and Performance Research Lab (EA 7425 HESPER) and EA 3738 CICLY, Université Lyon 1, Claude Bernard, 69921 Lyon, France; myriam.decaussin-petrucci@chu-lyon.fr (M.D.-P.); jean-christophe.lifante@chu-lyon.fr (J.-C.L.); 7Service d’Anatomopathologie, Hospices Civils de Lyon, Groupement Hospitalier Sud, 69495 Pierre Bénite, France; 8Centre de Médecine Nucléaire, Hospices Civils de Lyon, Groupement Hospitalier Est, 28, 69500 Bron, France; mireille.bertholon-gregoire@chu-lyon.fr (M.B.-G.); c.bully@lesportesdusud.fr (C.B.); agnes.perrin01@chu-lyon.fr (A.P.); 9Service de Chirurgie Viscérale et Endocrinienne, Hospices Civils de Lyon, Groupement Hospitalier Sud, 69495 Pierre Bénite, France

**Keywords:** papillary thyroid carcinoma, microscopic extra-thyroidal extension

## Abstract

**Simple Summary:**

Microscopic extra-thyroidal extension (mETE) in papillary thyroid carcinoma (PTC) has shown no effect on survival but controversy remains regarding its impact on disease recurrence. As it was removed from the American Joint Committee on Cancer (AJCC) Tumor-Node-Metastasis (TNM) classification in 2017, PTC tumors previously classified as pT3 are now downstaged to pT1 and pT2 tumors. This might imply that such tumors now need less aggressive treatments, especially complementary radioactive iodine treatment. We ought to assess if mETE remains a poor prognosis risk factor that still needs to be taken into account in the post-operative treatment strategy.

**Abstract:**

Background: This study assessed the risk of reduced disease-free survival (DFS) and poor clinical outcome in patients with papillary thyroid carcinomas (PTC) with microscopic extra-thyroidal extension (mETE), as compared to PTC patients without mETE. Methods: Retrospective analysis of a prospective database of patients treated by total thyroidectomy and radioactive iodine (RAI) with a five-year follow-up and tumors < 40 mm. In total, 303 patients were analyzed: 30.7% presented tumors with mETE, and 69.3% without. mETE was defined as extra-thyroidal invasion without skeletal muscle involvement. The primary outcome, DFS, was defined as the interval between initial treatment and any subsequent PTC-related treatment. The second outcome was the clinical status at five years. Results: In univariate analyses, the five-year DFS was significantly lower for tumors with mETE (62.4% versus 88.1%, *p* < 0.001). In multivariate analysis, mETE and massive lymph node involvement (LNI) were independent prognostic factors, associated respectively with a hazard ratio of 2.55 (95% CI 1.48–4.40) and 8.94 (95% CI 4.92–16.26). mETE was significantly associated with a pejorative clinical outcome at five years, i.e., biochemical/indeterminate response and structural persistence (Respectively OR 1.83 (95% CI 0.83; 4.06) and OR 4.92 (95% CI 1.87; 12.97)). Conclusion: Our results suggest that mETE is an independent poor prognosis factor of reduced DFS and predictive of poor clinical outcome.

## 1. Introduction

Papillary thyroid carcinoma (PTC) is a differentiated carcinoma usually known to have a low mortality rate and an indolent clinical course [1,2]. Clinicopathological features, such as advanced age, male sex, increased tumor size, multifocality, loco regional extension or metastasis, and extrathyroidal extension, are associated with poor prognosis [3]. More recently, several molecular alterations (e.g., *BRAF* V600E and *TERT* promoter mutations) have also been shown to be associated with poor clinical outcome and are now included in prognostic classification systems [4,5,6]. Extrathyroidal extension of papillary thyroid carcinoma is classified either as minimal extrathyroidal extension (mETE) or as gross extra thyroidal extension (gETE). Whereas gETE is identified as a poor prognostic factor, with an impact on both mortality and recurrence, mETE has not been shown to affect overall survival [7,8,9]. This, in part, has led to the 8th edition (2017) of the American Joint Committee on Cancer (AJCC) Tumor-Node-Metastasis (TNM) classification no longer considering the presence of mETE for the definition of pT3 tumors [4]. Therefore, tumors ≤ 40 mm in diameter with mETE are now classified as pT1 or pT2 rather than pT3. In parallel, there has been a decreased use of radioactive iodine (RAI) in the past years and in the most recent American Thyroid Association (ATA) guidelines, the use of post-operative RAI is no longer mandatory for low-risk patients (including pT1 pT2) [10]. Thus, patients with small tumors and mETE now considered pT1 or pT2 may not require radioiodine ablation. However, the association of mETE with disease recurrence remains controversial. For instance, although in a recent meta-analysis mETE was associated with an increased risk of recurrence, this was not the conclusion of several of the studies included wherein the authors suggest that these discrepant results may be partially explained by the different definitions of mETE itself [11].

In light of the unclear risk of recurrence associated with mETE, we conducted a study among patients with tumors ≤40 mm to evaluate the impact of mETE on five-year disease-free survival (DFS) as a proxy for recurrence and using a standardized definition of mETE (8th edition of the TNM).

## 2. Materials and Methods

### 2.1. Study Group

Patients admitted for RAI ablation within the year following total thyroidectomy, between 1 January 1998 and 30 June 2013, were retrospectively collected from the prospective database of the Department of Nuclear Medicine of the University Hospital of Lyon (France). The inclusion criteria were patients aged > 18 years at the time of surgery, with tumors ≤ 40 mm in size, with or without lymph node involvement (LNI), with uni- or multifocal PTC, with available data of ≥5 years after initial treatment. Patients with gETE, as well as those with unavailable pathology data or insufficient details to determine the extent of ETE, were excluded.

Baseline demographic data (age, sex) and pathology data (histology, tumor size, margin status, LNI, presence or absence of mETE) were collected. All pathology reports were analyzed, and tumors were restaged if justified. Patients with tumors that had a proportion of tall cells > 30%, or poorly differentiated cells > 20%, were considered has having aggressive pathological variants. Extents of surgical treatment along with RAI ablation were collected as well. Patients were divided in two groups based on the presence or absence of mETE; this was defined according to the 8th AJCC TNM edition as extra-capsular invasion without strap muscle involvement [4]. Patients had RAI activities ranging from 30 to 100 millicuries (mCi).

### 2.2. Follow-Up

Patients were followed each year by clinical examination and measurement of thyroglobulin and thyroglobulin antibodies after stimulation or not. Neck ultrasound (US) was performed <2 years following surgery, and at 5 years of follow-up. Additional imaging (computerized tomography (CT), fluorodeoxyglucose-positron emission tomography (FDG -PET), 131-I whole body scan…) was performed if necessary. The primary outcome was DFS defined as the interval between initial treatment, namely total thyroidectomy and post-operative RAI ablation, and any subsequent PTC-related treatment for recurrent or persistent disease: second RAI ablation or local neck surgery, but also surgery of metastasis. Patients with only biochemical abnormalities, i.e., detectable thyroglobulin levels without structural disease, were not considered as recurrent patients if no treatment was required. Follow-up of treatment-free patients was censored after 5 years. Patients were restaged after 1 year and at 5 years of follow-up, according to the 2015 ATA guidelines [10]: they were considered as being in remission if they had undetectable basal (<0.2 ng/mL) or stimulated thyroglobulin (using rh-TSH; <1 ng/mL) and no evidence of disease at clinical examination and/or neck US; they were considered as having an indeterminate response if they had a basal thyroglobulin level between 0.2 and 1 ng/mL or stimulated thyroglobulin level between 1 and 10 ng/mL; they were considered as having a biochemical incomplete response if they had a detectable basal (>1 ng/mL) or stimulated (>10 ng/mL) thyroglobulin, and no evidence of disease; they were considered as having structural incomplete response if they had local or metastatic persistent disease.

As patients with indeterminate response and those with biochemical incomplete response have similar risk of recurrence (respectively 15–20% and 20% risk to develop structural disease according to the 2015 ATA guidelines [10]), they were grouped together in the same category: biochemical/indeterminate response for determination of the clinical status at 5 years.

### 2.3. Statistical Analysis

DFS was the primary outcome. In univariate analyses, the 5-year DFS probabilities (with standard error, SE) were estimated using the Kaplan–Meier estimator and compared using the Log-rank test. Multivariate modeling was performed fitting the Cox proportional hazards model.

The 5-year clinical status was the secondary outcome. The contribution of clinical covariates to 5-year clinical status was analyzed fitting nominal polytomous logistic regression models. This method allowed to analyze the association between clinical covariates and biochemical/indeterminate response or structural persistence, respectively, in comparison to remission considered as baseline.

The following variables were studied by univariate analysis: mETE (presence/absence), LNI (N0-Nx, N1a, N1b), aggressive pathological variants (Yes/No), margin resection involvement (R1/R0), pathological tumor size in mm (<=10, ]10; 20], ]20, 40]), and advanced age (under or above 55 years). Nested models were compared using likelihood ratio tests. In all statistical tests, *p* < 0.05 was considered as statistically significant.

## 3. Results

### 3.1. Baseline Characteristics

A total of 303 patients met the inclusion criteria. Of these patients, 93 (30.7%) presented tumors with mETE, and 210 (69.3%) presented tumors without mETE. There was no significant difference between those with and without mETE in terms of mean age, sex, aggressive pathology variants, and metastasis at diagnosis. The distribution of LNI was significantly different between the two groups (*p* < 0.001). LNI was more frequent in patients with mETE than in those without (Table 1).

### 3.2. Disease Free Survival

The overall five-year DFS was estimated to be 80.2% (SE: 2.3%). It was estimated to be 62.4% (SE: 5.0%) for tumors with mETE, and 88.1% (SE: 2.2%) for tumors without (Log-rank, *p* < 0.001; Figure 1A); it was estimated to be 88.8% (SE: 2.1%) for N0-Nx tumors, 76.6% (SE: 6.2%) for N1a tumors, and 25.0% (SE: 7.7%) for N1b tumors (*p* < 0.001; Figure 1B); 37.7% (17.1%) for aggressive pathological types and 81.4% (2.3%) in others (*p* < 0.001; Figure 1C); and 60.0% (12.6%) in R1 tumors, and 81.3% (2.3%) in R0 tumors (*p* = 0.048; Figure 1D). Neither advanced age (*p* = 0.125) nor tumor size (*p* = 0.890) were significantly associated with prognosis.

In multivariate analysis, the presence of mETE was significantly associated with subsequent treatment (HR: 2.55, 95% CI [1.48; 4.40]), as were N1a and N1b tumors as compared to N0/Nx tumors (HR: 1.67, 95% CI [0.81; 3.46] and 8.94 (95% CI [4.92; 16.26]), respectively; Table 2), without interaction between mETE and LNI (*p* = 0.205).

### 3.3. Clinical Outcome

One to two years after initial treatment, 73.2% (*n* = 219) of the total population was considered in remission (53.2% of the mETE group and 82.1% of the non-mETE group), 12.0% (*n* = 36) in biochemical/indeterminate response (19.6% of the mETE group and 8.7% of the non-mETE group), and 12.7% (*n* = 38) in structural incomplete response (27.2% of the mETE group and 6.3% of the non-mETE group). Among the 219 patients considered in remission, three experienced a recurrence of disease within five years of follow-up: one had a second RAI administration because of rising thyroglobulin titer, one had neck surgery and second RAI administration because of local recurrence, and one had a second RAI administration and radiotherapy because of bone metastasis. The latter patient had a papillary carcinoma with aggressive pathology (>30% of poorly differentiated cells) and no mETE. Among the patients not considered in remission, 20 had local recurrent or persistent disease, and nine had pulmonary or bone metastasis. 

At five years of follow-up, 80.6% of the total population (*n* = 241) was considered in remission (65.6% of the mETE group, 87.3% of the non mETE group), 11.4% (*n* = 34) in biochemical/indeterminate response (17.2% of the mETE group, 8.7% of the non mETE group), and 8.0% (*n* = 24) in structural incomplete response (5.4% of the mETE group, 3.9% of the non mETE group). Between initial treatment and five-year follow-up evaluation, a total of 97 new treatments were recorded in the mETE group (range: 0–7). Nearly three-quarters (74.2%) of these were iterative RAI ablations, 19.6% were local neck surgeries, and the remaining were radiotherapy (2.1%) and metastasis surgeries (4.1%).

In the group without mETE, a total of 55 new treatments were recorded between initial treatment and five-year follow-up evaluation (range: 0–9). Two-thirds of these (72.7%) were iterative RAI ablations, 16.4% were local neck surgeries, and the remaining were radiotherapy (5.5%), metastasis surgeries (3.6%), and radiofrequency metastasis ablations for one patient.

In univariate analysis, mETE, aggressive pathology, and LNI were significantly associated with clinical status at five years; advanced age and tumor size were not, as was margin resection but there were only a total of five patients concerned (biochemical/indeterminate response, *n* = 2; structural persistence *n* = 3, Table 3). 

In multivariate analysis, mETE and aggressive pathology were significantly associated with structural persistence. Massive LNI (N1b) was significantly associated with biochemical/indeterminate response and structural persistence. Large effect-size point estimates were observed for tumors with massive LNI (N1b) and biochemical/indeterminate response (OR 7.28 [2.72; 19.48]), and for tumors with mETE and structural persistence (OR 4.92 [1.87; 12.97]). Aggressive pathology was significantly associated with a poor prognosis despite the limited number of patients (*n* = 8; Table 4).

## 4. Discussion

Microscopic extra thyroid extension is no more taken into account in the latest TNM classification and therefore no longer has an impact neither on T category nor overall stage [4], leading to the downstaging of tumors. This may therefore have consequences on patient management. Since conflicting results have been reported in the literature, we performed this study in order to assess if mETE has a prognostic value, independently from other usual criteria: tumor size, LNI, margin resection, presence of aggressive histologic variants. 

In our study, the distribution of patients with and without mETE was concordant with literature [12] with 93 (30.7%) patients presenting tumors with mETE, and 210 (69.3%) presenting tumors without mETE.

In this large series of carefully evaluated patients, mETE was an independent risk factor of poor clinical outcome in both the primary and secondary outcome. The presence of mETE was associated with a significantly lower DFS and was predictive of poor clinical outcome at five years. 

Previous studies attempting to assess the prognostic impact of mETE have shown discrepant results. The meta-analysis from Diker-Cohen et al., conducted on thirteen retrospective studies including 23816 patients, has suggested an increased risk of recurrence in papillary thyroid carcinoma with mETE [11]. Tran et al. performed a retrospective analysis of DFS in 577 patients, in which microscopic ETE was associated with multiple other adverse prognostic features and reduced DFS [13]. Jingzhe Xiang et al. found that mETE was significantly associated with poorer outcome on cancer-specific survival [14]. Parvathareddy et al. analyzed a cohort of 1430 patients and suggested that m-ETE is an independent marker for poor recurrence-free-survival (RFS) [15]. On the contrary, Nixon et al. [16] studied 984 patients and found no significant difference in 10-year disease-specific survival or RFS between the two groups. Hay et al. also found no difference in RFS between patients with mETE and no ETE [7]. More recently, Giorgio Grani et al. conducted a retrospective study that failed to show the prognostic value of mETE in predicting initial therapy response, in a cohort of PTC patients without LNI, and the retrospective series of Weber et al. found that mETE did not increase the risk for metastases at initial diagnosis and the recurrence rate [17,18]. In this last study, however, patients with mETE tended to be treated more aggressively: external beam radiation therapy was used more often higher median cumulative activities of RAI were used. Moreover, recurrence was defined as the detection of recurring structural disease by any imaging modality.

We chose DFS rather than RFS as the primary outcome and defined it as the need for additional treatment after initial therapy. Indeed, on one hand, even in situations in which remission cannot be asserted, treatment may not be justified, as the situation may remain stable for years: for instance in case of low detectable thyroglobulin titers (i.e., basal thyroglobulin level between 0.2 and 1 ng/mL or stimulated thyroglobulin level between 1 and 10 ng/mL) without US abnormalities. In this situation of indeterminate response, the 2015 ATA guidelines defined the disease specific death as lower as 1%, and the progression rate over years as lower as 20%. Moreover, the notion of restaging is now largely demonstrated and even patients with initial high-risk disease have a favorable long-term outcome if a complete response to initial treatment is obtained [10]. This is concordant with the results of our study: among the 219 patients considered in remission at one year of follow-up, only three had an actual local recurrence or rising thyroglobulin level; one of them had a PTC with aggressive histologic variant (>30% of poorly differentiated cells). On the other hand, situations in which structural recurrence is not certain still lead to additional treatment: in the case of biochemical persistence (i.e., Negative imaging and suppressed thyroglobulin >1 ng/mL or stimulated thyroglobulin >10 ng/mL or rising anti- thyroglobulin antibody levels), the 2015 ATA guidelines suggest that about 20% of these patients need additional therapies to achieve remission. Therefore, the definition of our primary outcome itself explains that the rate of DFS in the mETE group in univariate analysis is lower as compared to the rate of RFS in literature [18]. This is also explained by the higher rate of patients with LNI in the mETE group in our study. Indeed, other factors than mETE can affect DFS and especially the presence of LNI, which is a recognized factor of recurrence [19]. In his recent study, Parvathareddy et al. found that mETE was associated with LNI at the beginning of the disease, but also with distant metastasis [15]. However, despite the difference in the rate of patients with LNI between the two groups, mETE kept its own prognostic impact on DFS after adjustment on the other prognostic variables, as well as in our study.

In total, more treatments were needed in the mETE group, with a total of 97 treatments, versus 60 in the group without mETE.

Concerning clinical status at five years, the presence of mETE was significantly associated with unfavorable clinical status, i.e., structural persistence (In multivariate analysis, OR 4.92 [1.87; 12.97]), regardless of the presence of other significant adverse features (LNI and aggressive histologic variants). There was little change in point estimations between univariate and multivariate analysis. Indeed, concerning second outcome, mETE was associated with an OR of 5.90 [2.41–14.47] in univariate analysis and an OR of 4.92 [1.87–12.97] in multivariate analysis. This reinforces the status of mETE as an independent risk factor of poor clinical outcome.

The major differences in the definition of mETE Itself may also contribute to explain the heterogeneities among previous studies on the prognostic impact of mETE, as illustrated in the meta-analysis of Diker-Cohen [11]. Indeed, several studies defined mETE as an extension not seen by surgeons [7,16,20]. Others used sternothyroid muscle and peri-thyroid tissue invasion as a definition [9,21,22,23]. Some used the T staging report or did not mention which definition of mETE was used [12]. The definition of “minimal extension” has actually been changed between the 7th and the 8th editions: before considered as invasion of perithyroid tissue and/or sternothyroid muscle, it is now defined as extra-capsular invasion without strap muscle involvement [4]. Indeed, invasion of the strap muscle may be related to larger tumor size and worse RFS [24]. In our study, we used a standardized definition of mETE based on the latest TNM definition. This new definition can be controverted especially concerning isthmus located tumors, where strap muscle is closely intricated with thyroid tissue, making strap muscle involvement more common [25]. Out of all patients excluded because of strap muscle invasion, none had isthmus located tumors. Nevertheless, since the delineation of the extent of ETE is crucial, the absence of a systematic and independent review of the tumor slides by expert pathologists is a weakness of this study. To mitigate this point of weakness, we have eliminated all cases with insufficient data about ETE on pathologic reports.

Furthermore, most studies report no significant impact of mETE on recurrence rate in the specific case of micro carcinoma [7,12,21]. Other studies reported no impact of mETE on disease outcome in tumors below 15 mm, or after adjustment on tumor size, suggesting that tumor size is an important factor to take into consideration in the presence of mETE to assess the risk of recurrence [22,26]. Our study does not allow us to assess the impact of mETE adjusted on tumor size as we did not perform subgroup analysis. Surprisingly, the mean tumor size was significantly smaller in the mETE group (16.28 mm vs. 19.11 mm). However, this small difference has no clinical impact as T staging would be identical. On the contrary, Moon et al. reported larger tumors in the cohort of patients with mETE as compared with patients without [21].

Previous studies had heterogeneous populations concerning initial treatment. All our patients were treated with total thyroidectomy and post-operative RAI ablation, which allowed us to have a homogeneous population, but not to assess the evolution of tumors with mETE in the absence of RAI ablation. Further guidelines should take the new definition of mETE into account. British guidelines of 2014 consider extra thyroidal extension as an uncertain indication of post-operative RAI ablation, in the absence of LNI or other unfavorable features [27]. A more recent Italian consensus suggests that in case of extra thyroidal extension, thyroidectomy should be preferred over hemi-thyroidectomy, but this extra thyroidal extension is defined as extension seen during ultrasound examination. According to these guidelines, the use of complementary RAI ablation should be decided on the basis of both the AJCC staging and the Initial Risk Stratification System proposed by ATA [28]. French guidelines propose that mETE might be taken into account in the decision of post-operative RAI ablation, along with other adverse features [29]. Lately, the 2021 European Thyroid Association Guidelines on indications for post-surgical radioiodine therapy in differentiated thyroid cancer still consider small tumors with minimal extension as intermediate risk tumors, suggesting that the benefit of RAI therapy is controversial but should be discussed, based on post-operative thyroglobuline and neck ultrasound [30].

This study is limited by its retrospective nature. Prospective studies should be realized to bring a definite answer to this problem and should address the question of the optimal treatment of these patients, especially regarding the post-operative radioiodine administration. It could be interesting to proceed to other studies taking molecular markers into account, in order to estimate if mETE remains associated with prognosis in models including molecular markers, such as *BRAF* V600E or *TERT* promoter mutations [15,31]. In such situations, oncogenic alterations may help in tailoring treatment decision making.

## 5. Conclusions

The results of our study, combined with the results of Diker-Cohen’s meta-analysis, suggest that mETE should still be taken into account in the post-operative strategy, even if it has been removed from the current TNM classification.

## Figures and Tables

**Figure 1 cancers-14-02591-f001:**
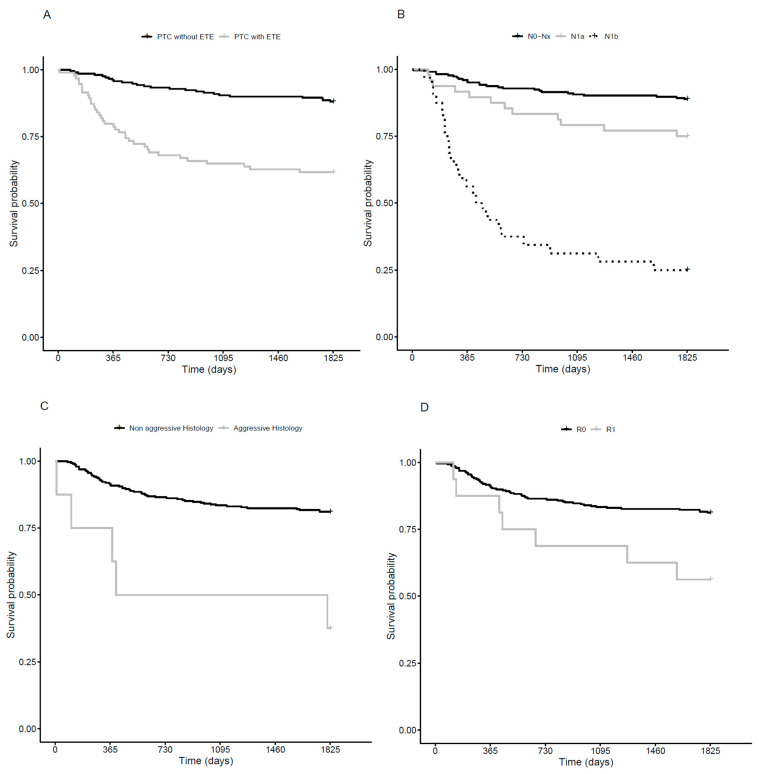
Univariate analysis (Kaplan-Meier analysis). (**A**). Recurrence-free survival (RFS) in patients with mETE vs. no mETE. (**B**). RFS in patients with aggressive histology vs. no aggressive histology carcinoma. (**C**). RFS in patients with pathological margin resection (R1) vs. no pathological margin resection. (**D**). RFS in patients with LNI (N1b or N1a) vs. no LNI (N0-Nx).

**Table 1 cancers-14-02591-t001:** Patient characteristics.

		mETE (*n* = 93)	No mETE (*n* = 210)	*p* Value
Mean age, years (range)		57 (25–86)	57 (29–90)	0.695
Female, *n* (%)		73 (78.5)	170 (80.9)	0.620
Aggressive pathology, *n* (%)		5 (5.3)	3 (1.4)	0.061
Lymph node involvement, *n* (%)				<0.001
N0/Nx		51 (54.8)	173 (82.3)	
N1a		24 (25.8)	23 (11.0)	
N1b		18 (19.4)	14 (6.7)	
Margin resection, *n* (%)	R0 R1	80 (86) 13 (13.9)	208 (99) 2.0 (0.9)	<0.001
Tumor size, mm	Mean Range	16.3 3–39	19.1 1–39	0.006
Metastasis at diagnosis, *n* (%)		6 (6.4)	6 (2.8)	0.198

**Table 2 cancers-14-02591-t002:** Multivariate analysis of factors associated with a subsequent treatment.

		Hazard Ratio	[95% CI]	*p* Value
mETE	No Yes	1 2.55	[1.48; 4.40]	<0.001
Lymph node involvement	N0/Nx N1a N1b	1 1.67 8.94	[0.81; 3.46] [4.92; 16.26]	<0.001

**Table 3 cancers-14-02591-t003:** Univariate analysis of factors associated with clinical outcome at 5 years.

		Rem. (*n*)	Bio/In (*n*)	Struct (*n*)	Bio/In (OR)	Struct (OR [95% CI])	*p* Value *
Age, years	≤55 >55	168 73	24 10	16 8	1 0.96 [0.44; 2.11]	1 1.15 [0.47; 2.81]	0.945
mETE	No Yes	180 61	18 16	8 16	1 2.62 [1.26; 5.46]	1 5.90 [2.41; 14.47]	<0.001
Aggressive pathology	No Yes	238 3	33 1	20 4	1 2.40 [0.24; 23.8]	1 15.87 [3.32; 75.88]	0.005
Lymph node involvement	N0/Nx N1a N1b	192 35 14	16 8 10	14 3 7	1 2.74 [1.09; 6.90] 8.57 [3.29; 22.35]	1 1.18 [0.32; 4.30] 6.86 [2.38; 19.74]	<0.001
Margin resection	R0 R1	231 10	32 2	21 3	1 1.44 [0.30; 6.89]	1 3.30 [0.84; 12.43]	0.289
Tumor size, mm	≤10 10–20 >20	41 105 95	9 15 10	4 12 8	1 0.48 [0.18; 1.27] 0.65 [0.26; 1.60]	1 0.86 [0.25; 3.03] 1.17 [0.36; 1.60]	0.641

Rem.: Remission. Bio/In.: Biochemical/Indeterminate Struct.: Structural persistence. OR: Odds Ratio. * likelyhood ratio test.

**Table 4 cancers-14-02591-t004:** Multivariate analysis of factors associated with clinical outcome at 5 years.

		Rem. (*n*)	Bio/In. (*n*)	Struct. (*n*)	Bio/In (OR)	Struct. (OR)	*p* Value *
mETE	No Yes	180 61	18 16	8 16	1 1.83 [0.83; 4.06]	1 4.92 [1.87; 12.97]	0.003
Aggressive pathology	No Yes	238 3	33 1	20 4	1 1.90 [0.17; 20.86]	1 12.84 [2.23; 73.97]	0.020
Lymph node involvement	N0/Nx N1a N1b	192 35 14	6 8 10	14 3 7	1 2.27 [0.87; 5.92] 7.28 [2.72; 19.48]	1 0.59 [0.14; 2.48] 3.98 [1.23; 12.84 ]	0.001

Rem.: Remission. Bio/In.: Biochemical/Indeterminate. Struct.: Structural persistence. OR: Odds Ratio. * likelyhood ratio test.

## Data Availability

The data presented in this study are available on request from the corresponding author.

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
