# Peer review of "Prognostic Impact of Microscopic Extra-Thyroidal Extension (mETE) on Disease Free Survival in Patients with Papillary Thyroid Carcinoma (PTC)"

_cancers, 2022, doi:10.3390/cancers14112591_

Round 1

Reviewer 1 Report

Dear Authors

General Comment: I read the manuscript with interest. The topic is of great interest, retrospective on a court of 303 patients with papillary thyroid cancer. The authors of the role of their own experience with respect to EXTRA-THY-2 ROIDAL EXTENSION (goals) on disease-free survival in patients with papillary thyroid cancer. The prognostic role of MICROSCOPIC EXTRA-THY-2 ROIDAL EXTENSION has long been a subject of controversy in the literature.

My main comments are as follows:

  1. Check line 91
  2. Both the objectives and methods of this research are broad and radical
  3. The images are of good quality and standardized.
  4. Thyroid part needs to be improved, its relationship part needs to be added to the understanding of the molecular pathogenesis underlying the progression of thyroid cancer, particularly for papillary thyroid carcinoma (PTC) and its roles on downstaging and prognosis for the thyroid gland make your study interesting.

I recommend reading these papers in references: 

Ulysses S et al. Papillary Thyroid Cancer Prognosis: An Evolving Field Cancers 2021, 13 (21), 5567

Sorrenti S et al. Evaluation of clinicopathological and molecular parameters on disease recurrence of the patient with papillary thyroid carcinoma: a retrospective observational study Cancers 2020, 12 (12), pp. 1-13, 3637

Xing, Mingzhao. Genetically Guided Risk Assessment and Management of Thyroid Cancer Endocrinology and Metabolism Clinics of North America 2019 1, pages 109 - 124

- Clarify all abbreviations used in the text

-The images are of good quality and standardized.

Reviewer 2 Report

The manuscript presents some interesting and possibly very useful information which has the potential to influence the staging and management of thyroid cancers.  That being said, there are a few significant issues which should be addressed.

1) It appears that the authors did not actually review the pathology slides and determine in a uniform manner the presence or absence of mETE which creates a problem since mETE can be subjective in some cases.  Given the tremendous importance placed upon mETE for this study, the authors themselves must assess this trait rather than simply taking it from a pathology report.  This study would be much stronger if there were a uniform review and unanimous consensus among 2-3 study pathologists using standard, agreed upon criteria for the determination of mETE.

2) The extent of mETE may influence the impact of this feature.  Is the mETE 0.01 cm or 0.1 cm or 0.2 cm?  Evaluating and stratifying the actual degree of mETE could be very useful since not all mETE may be the same!

3)The word "pejorative" is not typically used in English to describe clinical outcome, and may be confusing to the reader.
